# Use of Social Networks in University Studies: A Peruvian Case Study



Nataly Puma-Chavez [1], Jheidys Quispe-Escobar [1], Alejandra Hurtado-Mazeyra [1] and Carmen Llorente Cejudo [2,*]

1   Academic Department of Education, National University of San Agustín de Arequipa, Santa Catalina Street No. 117, Arequipa 04001, Peru
2   Department of Teaching and Educational Organization, University of Seville, 41013 Seville, Spain
*   Correspondence: karen@us.es

**Abstract:** The aim of this study was to determine the degree of addiction to social networks among students at a Peruvian university. The sample consisted of 3026 students (1768 females, 58.4%; 1258 males, 41.6%) from three different fields of study: social sciences, engineering, and biomedical sciences. Data were collected using the Social Media Addiction Scale–Student Form (SMA–SF), which was previously adapted to the Peruvian context, establishing four dimensions: satisfaction/tolerance, problems, obsession with being informed and need to be connected. Among the results obtained, it is possible to highlight factors that influence the use of social networks, such as, that their use varied according to age, year of university studies, gender, and area of study; in addition, significant differences in gender were observed, with men using them more for various activities and expressing feelings about their use. Engineering and Social Science students were more likely to show feelings towards networks than Biomedical Science students, among others. Students in the 16–20 age range showed greater expressions of feelings about the use of social networks than those in other age ranges.

**Keywords:** social networks; addiction; university students

## 1. Introduction

It is important to mention the remarkable impact that social networks have made in recent years, and it cannot be denied that the millions of users worldwide are unquestionable. A recent statistical study [1] indicated that, by the year 2021, the number of Internet users had increased to 4.9 billion, noting that Facebook continued to lead the world ranking, as more than 1 in 3 people accessing the Internet worldwide (35.85%) visit this website, the second most-visited social network or website in the world, after Google [2]. Others also pointed out that Twitter and Instagram were the platforms with the highest adoption and frequency of use among students [3–6].

In different continents, social networks and internet use has increased, so much so that in Asia the level of internet penetration is above 70% (e.g., Japan: 91.1%, South Korea: 85.7%, Singapore: 82.5% and Hong Kong (HK)/Macao: 74.1%) [7]; in countries such as Iran, the use of social networks has tripled in the last three years [8], with Facebook being the most popular social network in some cases, such as in Pakistan, with 8 million users [9]. In Europe, Internet users account for 85% of all 16–74 year olds and 97% of all 16–25 year olds. Among these Internet users, 65% use social networks [10]; Spain, for example, was 50 points above the world peak in web traffic, due to the increase in the number of social network users [11]. In the Americas, there are an estimated 673.1 million users, indicating the expansion of social networks in North America (70%), Central America (62%), the Caribbean (42%) and South America (66%), with users aged between 25 and 34 [12]. In Peru, during the third quarter of 2021, domestic internet access reached 55%, an increase of 9.6 percentage points [13]. It is therefore clear that the massive diffusion of social networks and internet connection are generating new behaviours in individuals and society [14].

Social networks are defined as interaction systems that people use with friends/ acquaintances ("contacts"), with whom they interact through an electronic platform associated with the Internet, such as Facebook, Twitter and Instagram, among others [15]. The concept of social networks today implies a new way of understanding, seeing and perceiving communication between individuals, with as many critics as followers [16]. Although all definitions of digital social networks (DSNs) are varied, they all assume that they constitute a space in which individuals interact, share information and communicate with each other to create communities; this can be considered a network [17].

It is evident that the use of social networks is more frequent in the adolescent and young populations [17–20]. They are the ones who most frequently incorporate social networks and new technologies into their daily lives for the exchange of information and leisure [3,21], acting as protagonists and assuming the role of consumers and generators of content [22]. This is the case of applications such as TikTok, which was designed to create and share short videos [23], making this social network easy to use, even for the youngest users. Thus, social networks are a medium that provides proximity and interaction between Internet users, and allow for interpersonal relationships at a distance [18,24,25], as they provide the opportunity to establish relationships for those who have difficulties with face-to-face interaction [20,26].

Social networking and video games are also closely related [6,27]. Thus, more and more people are starting to use the Internet, and online games have become extremely popular among the younger generation [28], allowing users to have fun while escaping from reality.

Moreover, in recent years, the use of social networks in education has increased, with benefits reported by different authors. They allow communication to be established, as they are perceived as useful and friendly, increasing the fluidity and simplicity of communication between teachers and students [29,30]. Different studies have revealed that Facebook is one of the alternative means for shy students to express their ideas inside and outside the classroom, as it allows them to collect and present them in writing instead of verbally [31]. Social networks are meaningful learning environments characterised by participation and interaction within organisations; they are interactive learning environments centred on the active role of the learner, who takes part in the educational responsibility [31,32]. They favour significant attitudes towards collaborative work and the collective construction of knowledge [17,33]. They are highly motivating environments for students [34], improve self-esteem and allow students to perceive greater social acceptance by other students [35]. They influence academic performance [30], improving memory management, with better results in cognitive language skills tests [18]. Finally, they facilitate immediate feedback for students.

Although there are many benefits, the excessive use of social networks triggers a series of negative effects, including social network addiction. A review of the literature on social network addiction shows that different authors associate it with a behaviour that makes it difficult for individuals to perform normally in their daily lives, becoming dependent on social networks, presenting the following symptoms: (1) loss of self-control; (2) intense desire to connect to social networks; (3) signs of withdrawal (e.g., anxiety, nervousness, depression and irritability) when not accessing social networks; (4) tolerance (need to stay online longer to obtain the same results and feel satisfied); (5) severe interference in daily life; and (6) progressive abandonment of other activities that generated pleasure [11,20,36,37]. In addition, other authors mention that the term addiction is only used when referring to the abusive use of drugs and chemicals [38], and clarify that addiction is defined as the abuse of and dependence on any psychoactive substance [39]. Although there is no consensus on the use of the concept of "social network addiction", numerous studies have addressed this issue, which particularly affects young people worldwide.

Different studies have contributed to the study of social network addiction or abuse. For example, those that highlight the high frequency of problems associated with the excessive use of social networks, such as difficulty in socializing and controlling one's

emotions [40–42]. Others associate it with an unhealthy strategy for coping with negative moods and daily problems [43,44]; in other cases, a high degree of addiction to social networks is reported, with Instagram being the social network most used by students to communicate [36].

## 2. Materials and Methods

The aim of this exploratory study was to determine the degree of addiction of students at the National University of San Agustin de Arequipa (Peru), using the adapted scale [45], known as "Social Media Addiction Scale–Student Form" (SMAS–SF). Additionally, we analyzed the existence of significant differences regarding the degree of addiction to social networks as a function of the following variables: gender (O1), year of the university degree (O2), study area (O3) and age (O4). The study did not require any reporting or the establishment of an ethics committee.

### 2.1. Study Population and Sample

The population of this study was constituted by students from the National University of San Agustin de Arequipa (Peru). A probabilistic and purposeful sampling was conducted based on the requirements of the study, considering volunteer individuals from the target study population [46], using non-random methods to select a sample whose characteristics were similar to those of the target population.

The sample consisted of 3026 students at the National University of San Agustin de Arequipa from three study areas (engineering, biomedical sciences and social sciences), with 1768 women (58.4%) and 1258 men (41.6%) (Table 1).

**Table 1.** Distribution of the sample according to student gender.

| Students of the Sample | F | % |
|:---:|:---:|:---:|
| Men | 1258 | 41.6% |
| Women | 1768 | 58.4% |

With respect to the year of university degree, Figure 1 shows that most of the participants of the sample were in the first year (f = 791; 26.1%), whereas those in the sixth year represented the smallest group in the entire sample (f = 27; 0.9%).

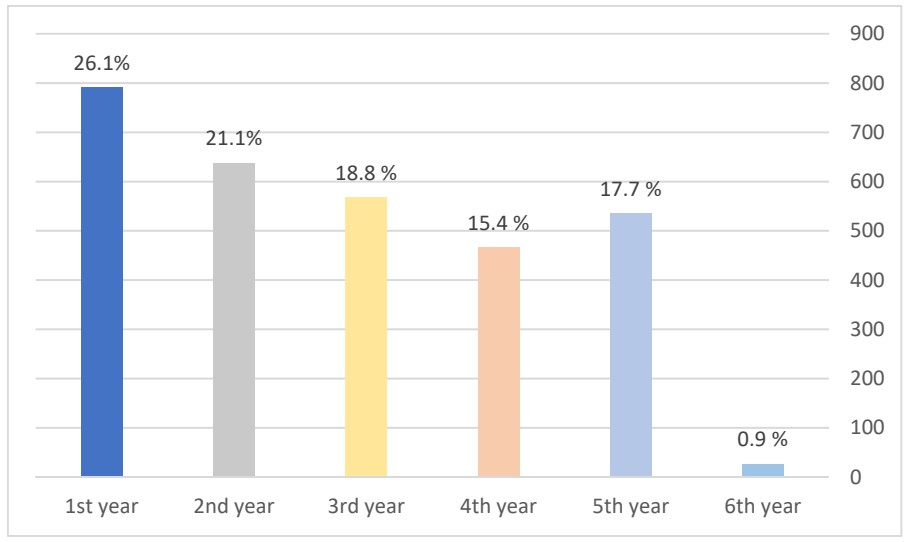

**Figure 1.** Year of university degree.

Table 2 presents the data of the students in relation to the study areas in which they were registered, with a predominance of social sciences and engineering (51.4% and 39.8%, respectively) over biomedical sciences (f = 269; 8.9%).

**Table 2.** Data of the participants' study area.

| Study Area | F | % |
|---|---|---|
| Social sciences | 1554 | 51.4% |
| Biomedical sciences | 269 | 8.9% |
| Engineering | 1203 | 39.8% |

Lastly, Figure 2 shows the age distribution of the participants; as can be observed, the students aged 16–20 years constituted the largest group of the entire sample in this respect (f = 1472; 48.6%), whereas those aged 25 years or older represented the smallest group.

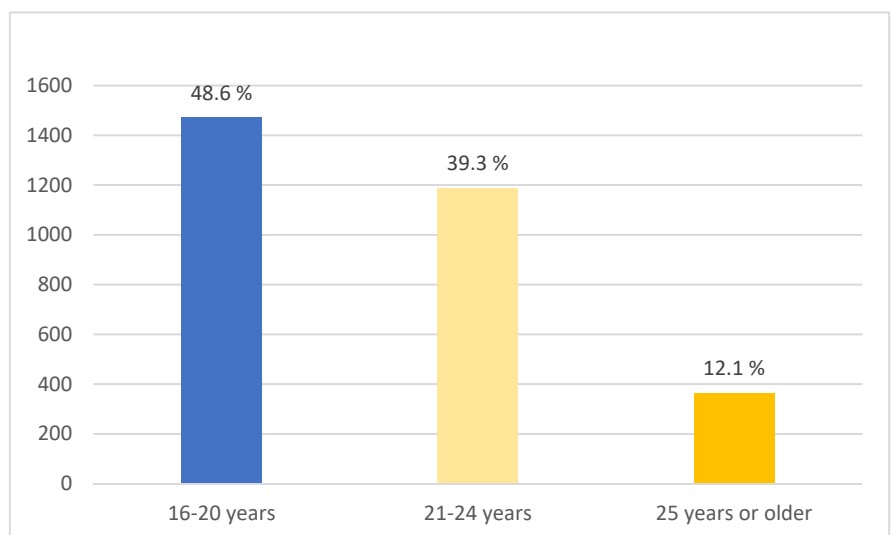

**Figure 2.** Age of the participants.

*2.2. Instrument*

The instrument has four factors that could explain the addiction to, or constant use of, social networks:

- A. Use of social networks (0 = little, 5 = much)
- B. Use the internet for activities (0 = little, 5 = much)
- C. Use of social networks in activities (0 = little, 5 = much)
- D. Usefulness of social networks (0 = little, 5 = much)

SMAS–SF was selected for its novelty, the theoretical basis provided by the author for its development and the robust process followed for its creation, which consisted of four major phases: (1) literature review, (2) construction of a Likert-type instrument, (3) validation through expert judgement, and (4) analysis of its construct reliability and validity.

The data obtained from the scale were processed using SPSS software v24 (Statistical Package for the Social Sciences) which is the most popular statistical package for data processing especially in the field of education. It uses descriptive statistics and non-parametric contrast statistics [47].

The questionnaire was administered online in the year 2021, ensuring the reliability of the answers by sending it to the university e-mail addresses of the respondents, informing the latter about the objectives of the study. All participants signed a confidentiality agreement and agreed to provide their data included in the first part of the questionnaire.

**3. Results**

The results obtained in this study are presented based on the following procedure:

- Presentation of the descriptive statistics obtained.
- Comparison of variables (gender, year of the university degree, study area, and age).

### 3.1. Descriptive Statistics

The results obtained in the administration of the scale provided different mean scores and standard deviations for each of the different items proposed. Firstly, we presented the global nominal descriptive data for the different variables analysed (see Table 3):

- Use of social networks
- I use the Internet for the following activities
- Use of social networks in activities
- Usefulness of social networks to

**Table 3.** Mean values and standard deviations of the different items for the global sample.

| | Mean | SD |
|---|---|---|
| **(A) Use of social networks** | | |
| A1. How frequently do you use social networks? | 3.54 | 0.908 |
| A2. How many hours do you spend using social networks every week? | 3.05 | 1.388 |
| A3. How many hours of the day do you normally use social networks? | 4.46 | 1.975 |
| **(B) I use the Internet for the following activities:** | | |
| B1. Social networks (Facebook, Instagram, Snapchat, Twitter, LinkedIn, etc.) | 3.79 | 1.218 |
| B2. Instant messaging | 3.17 | 1.236 |
| B3. Chats or groups (WhatsApp, Telegram, Viber, WeChat, Line, etc.) | 4.19 | 0.861 |
| B4. Freely browsing different websites | 3.76 | 0.922 |
| B5. Searching information that I need for my studies | 4.39 | 0.706 |
| B6. Searching and downloading different things (music, images, documents, etc.) | 3.88 | 0.948 |
| B7. Watching films or series directly, without downloading them | 3.00 | 1.207 |
| B8. Listening to music or watching videos, without downloading them | 3.65 | 1.073 |
| B9. Online games | 2.35 | 1.279 |
| **(C) Use of social networks in activities:** | | |
| C1. I can't wait to connect to some social network | 2.55 | 0.968 |
| C2. I search for Internet connection everywhere to access social networks | 2.44 | 1.026 |
| C3. The first thing I do when I wake up is connect to social networks | 2.89 | 1.135 |
| C4. I see social networks as a way to escape the real world | 2.40 | 1.126 |
| C5. A life without social networks has no meaning to me | 1.78 | 0.869 |
| C6. I'd rather use social networks even if I'm not alone | 1.97 | 0.906 |
| C7. I prefer friendships on social networks to face-to-face interaction | 1.76 | 0.905 |
| C8. I express myself better with people in social networks | 2.43 | 1.082 |
| C9. In social networks, I am whatever I want to show | 1.97 | 0.973 |
| C10. In general, I'd rather communicate with people through social networks | 2.30 | 0.997 |
| C11. Even my family is upset because I can't stop using social networks | 1.97 | 0.982 |
| C12. I want to spend time in social networks when I'm alone | 2.77 | 1.103 |
| C13. I'd rather communicate through social networks to arrange an outdoor activity | 2.67 | 1.134 |
| C14. The activities in social networks are fundamental in my daily life | 2.49 | 1.037 |
| C15. I omit my tasks because I spend too much time in social networks | 2.05 | 0.964 |
| C16. I feel unhappy when I'm not using social networks | 1.97 | 0.937 |
| C17. I get excited when I'm using social networks | 2.57 | 0.949 |
| C18. I use social networks so frequently that I forget about my family | 1.72 | 0.817 |
| C19. The mysterious world of social networks always captivates me | 2.44 | 1.013 |
| C20. I don't even feel hunger or thirst when I'm using social networks | 1.74 | 0.873 |
| C21. I feel that my productivity has decreased due to social networks | 2.56 | 1.149 |
| C22. I have physical problems due to the use of social networks | 2.13 | 1.014 |
| C23. I use social networks even when I walk down the street, so I'm instantly informed about events | 2.22 | 1.034 |
| C24. I like using social networks to be informed about what's happening | 3.41 | 1.015 |
| C25. I browse social networks to be informed about what the media groups are sharing | 3.16 | 1.046 |
| C26. I spend more time in social networks to see some special events (e.g., birthdays, parties, etc.) | 2.47 | 1.033 |
| C27. My need for being informed about things related to my studies (e.g., assignments, activities) makes me use social networks constantly | 3.65 | 1.088 |
| C28. I'm always active in social networks to be immediately informed about what my friends and relatives are sharing | 2.58 | 1.007 |

**Table 3.** *Cont.*

| | Mean | SD |
|---|---|---|
| **(D) Usefulness of social networks to:** | | |
| D1. Receive information | 3.81 | 1.996 |
| D2. Communicate with friends anDrelatives | 3.23 | 2.444 |
| D3. Study | 4.03 | 2.098 |
| D4. Meet new people | 2.51 | 2.962 |

The obtained data show that the items with the greatest score were D3 (usefulness of social networks) (8.03), D1 (usefulness of social networks) (7.81), D2 (usefulness of social networks) (7.23), and A3 (use of social networks) (4.46).

Thus, we can conclude that the students showed special interest for the use of social networks to study and learn, as well as to receive information and communicate with their friends and relatives.

Moreover, the hours in which they used social networks most frequently was between 2 p.m. and 4 p.m., indicating that students connect to social networks at hours in which they have free time to connect to the Internet.

On the other hand, the four items that obtained the lowest frequency score were the following: C18 (use of social networks) (1.72); C20 (use of social networks) (1.74); C7 (use of social networks) (1.76); and C5 (use of social networks) (1.78).

Among the presented items, several aspects are worth pointing out. For instance, the students showed that they used social networks several times per day, whereas a lower percentage of participants reported a frequency of once per day, several times per week, 2–3 times per week or at least once per week.

The students totally disagreed with item C18; thereby indicating that they do not forget about their relatives due to the use of social networks.

Likewise, the low score obtained in item C20 showed that the participants totally disagreed with this statement; in other words, they value social networks, but they do not allow them to influence their eating habits.

Similarly, the participants totally disagreed with item C16, thus, we can infer that using or not using social networks does not influence their mood.

Next, Table 4 presents the global mean values and standard deviations obtained for the students in the three dimensions of the addiction scale used.

**Table 4.** Mean scores and standard deviations for the global dimensions.

| | Mean | SD |
|---|---|---|
| (A) Use of social networks | 3.5757 | 0.59026 |
| (B) Use the internet for activities | 2.3956 | 0.57654 |
| (C) Use of social networks in activities | 2.5265 | 0.63254 |
| (D) Usefulness of social networks | 3.2568 | 1.64503 |

As can be observed in the obtained results, the students showed lower scores when asked about expressing their feelings in the use of social networks, whereas the highest scores are related to the usefulness of social networks in their lives, for both studying and leisure.

*3.2. Comparison of Variables*

It is important to highlight that the test of the hypothesis presented below was carried out with the total score obtained by the participants in the entire instrument used. To perform the analysis, the following non-parametric statistics were applied: Mann–Whitney U-test and Kruskal–Wallis H with post hoc test (Dunn's test).

It has been verified that the data are not normally distributed through a descriptive study in which skewness and kurtosis have been taken into account. The Kolmogorov–

Smirnov goodness-of-fit test confirmed this finding, with a significance (*p*-value) equal to 0.000 for all items (non-normal distribution)

Regarding the variable "gender", the results obtained in the Mann–Whitney U-test are presented in Table 5

**Table 5.** Correlation between gender and addiction to social networks.

| | Mann–Whitney U-Test | Wilcoxon's X | Z | Asymptotic Sig. |
|---|---|---|---|---|
| (A) Use of social networks | 1,035,075.000 | 2,598,871.000 | −3.256 | 0.001 |
| (B) Use the internet for activities | 1,012,682.000 | 2,576,478.000 | −4.197 | 0.000 |
| (C) Use of social networks in activities | 1,012,002.000 | 1,870,215.500 | −0.845 | 0.325 |
| (D) Usefulness of social networks | 1,089,122.500 | 1,881,033.500 | −0.970 | 0.332 |

The presented values allowed the rejection of the H0 formulated in the first two cases, which refers to the absence of significant differences as a function of gender when making use of the Internet for different activities and when expressing feelings in the use of social networks (≤0.05). Consequently, the alternative hypothesis (H1) is accepted, which refers to the existence of statistically significant differences in the use of the Internet for different activities, as well as in the expression of feelings in the use of social networks. The average range analysis shows statistically significant differences in favour of the female participants in both dimensions, as can be observed in Table 6.

**Table 6.** Average range analysis for the test variable "gender".

| | Gender | Average Range |
|---|---|---|
| (A) Use of social networks | Male | 1574.71 |
| | Female | 1469.95 |
| (B) Use the internet for activities | Male | 1592.51 |
| | Female | 1457.28 |

In this case, and considering the items of the variable "usefulness of social networks", Table 7 shows the values obtained for the acceptance of the alternative hypothesis, revealing the gender differences in the item "communicating with my friends and relatives". Therefore, these results suggest that social networks are used differently, as a function of gender, regarding communication with family members and friends.

**Table 7.** Gender differences with respect to the variable "D. Usefulness of social networks". Source: developed by author.

| C. Usefulness of Social Networks | Mann–Whitney U-Test | Wilcoxon's X | Z | Asymptotic Sig. (Bilateral) |
|---|---|---|---|---|
| 1. Receiving information | 1,042,269.000 | 1,834,180.000 | −3.003 | 0.003 |
| 2. Communicating with friends and relatives | 1,083,489.000 | 1,875,400.000 | −1.221 | 0.222 |
| 3. Studying and learning | 901,814.500 | 1,693,725.500 | −9.100 | 0.000 |
| 4. Meeting new people | 952,722.000 | 2,516,518.000 | −6.764 | 0.000 |

With respect to the year of university degree (O2), the main differences were established between the first-year and fourth-year students (Table 8).

**Table 8.** Correlation between the year of university degree and addiction to social networks.

|  | Kruskal–Wallis H | df | Asymptotic Sig. |
|---|---|---|---|
| (A) Use of social networks | 28.489 | 5 | 0.000 |
| (B) Use the internet for activities | 10.838 | 5 | 0.055 |
| (C) Use of social networks in activities | 5.255 | 5 | 0.215 |
| (D) Usefulness of social networks | 7.688 | 5 | 0.174 |

Thus, we can reject the null hypothesis formulated for the first case, which refers to the absence of significant differences between being in the first or fourth year and the use of the Internet for different activities. Consequently, the alternative hypothesis is accepted, which refers to the existence of statistically significant differences in this regard. However, it is important to maintain the null hypothesis in the second and third cases, which refers to the absence of statistically significant differences in the expression of feelings in the use of social networks and the usefulness of social networks, respectively.

The average range analysis shows statistically significant differences in favour of the sixth year (1733.63), whereas the least significant differences were detected for the first year (1423.13), with a gradual increase in the significance of the differences along the years of the university degree.

The third variable refers to the study area that the participants belong to (O3). In this case, as can be observed in Table 9, the data show that H0 can only be accepted in the first variable ("use of the Internet for different activities") and rejected in the other two variables.

**Table 9.** Correlation between the study area and addiction to social networks.

|  | Kruskal–Wallis H | df | Asymptotic Sig. |
|---|---|---|---|
| (A) Use of social networks | 3.545 | 2 | 0.170 |
| (B) Use the internet for activities | 14.633 | 2 | 0.001 |
| (C) Use of social networks in activities | 8.256 | 2 | 0.325 |
| (D) Usefulness of social networks | 7.357 | 2 | 0.025 |

Therefore, the values allow asserting that there are statistically significant differences in the expression of feelings in the use of social networks and in the usefulness of social networks. However, the use of the Internet for different activities did not show statistically significant differences between the study areas.

For this case, the average range analysis revealed significant differences in dimension B ("expression of feelings in the use of social networks") in favour of engineering (1554.83) and social sciences (1513.36), followed by biomedical sciences (1329.47). Similarly, regarding the usefulness of social networks, the results showed that these significant differences were in favour of biomedical sciences and social sciences (1557.10 and 1546.87, respectively).

It is important to highlight that age (O4) was the last variable on which the tests were performed, in different intervals: (a) 16–20 years, (b) 21–24 years, (c) 25 years or older. In this sense, we can assert that there were statistically significant differences in the use of the Internet for different activities, as well as in the usefulness of social networks, as can be observed in the results presented in Table 10.

**Table 10.** Correlation between age and addiction to social networks.

|  | Kruskal–Wallis H | df | Asymptotic Sig. |
|---|---|---|---|
| (A) Use of social networks | 0.497 | 2 | 0.780 |
| (B) Use the internet for activities | 8.168 | 2 | 0.017 |
| (C) Use of social networks in activities | 3.156 | 2 | 0.254 |
| (D) Usefulness of social networks | 2.591 | 2 | 0.274 |

The different results allow the rejection of H0 in the second case, which refers to the absence of statistically significant differences as a function of age in the expression of feelings in the use of social networks. Consequently, there are statistically significant differences in the expression of feelings in the use of social networks. Similarly, the results show that there are no statistically significant differences in dimensions A or C.

For this case, the average range analysis showed significant differences in dimension B ("expression of feelings in the use of social networks") in favour of the age group of 16–20 years, with an average range of 1555.45, whereas the age groups of 21–24 years and 25 years or older obtained an average range of 1489.08 and 1423.87, respectively).

## 4. Discussion and Conclusions

The aim of this study was to determine the degree of addiction to social networks in a population of university students at UNSA, as well as the existence of significant differences in the variables, such as gender, year of university studies, area of study and age.

In this sense, and in view of the results obtained and presented in this study, different conclusions have been drawn with respect to the proposed research questions.

Firstly, the UNSA students who participated in the study showed a special interest in the use of social networks for studying and learning, which coincided with the results of previous similar studies [20,39,48]. Likewise, the use of social networks was associated with actions such as receiving information and communicating with friends and family, which is relevant if we consider that this is their fundamental purpose [16,25,49].

It is worth noting that, due to the social and playful use of social networks, it is understandable that the preferred and most frequent time of use among the participants was between 14:00 and 16:00, which may be due to different reasons, although it can be inferred that this is the time when students finish their face-to-face or online classes and have more free time to connect to social networks and participate in them, while in the afternoon they return to study or carry out other activities, a fact that was repeated in other similar studies [50].

It is important to point out the preconceptions that young people have about the use of social networks, and one of the conclusions of the study is that they state that there are elements that are more relevant to them, such as spending time with their families or having good eating and sleeping habits. In this sense, it would be advisable to carry out a more in-depth, qualitative analysis to find out the origin of these manifestations, including the involvement of determining agents, such as family and teachers.

To facilitate the presentation of the conclusions, we present some of the conclusions we have reached, considering the proposed research questions. Thus, it can be affirmed that there are differences with respect to gender, in terms of the use that UNSA students make of social networks and, more specifically, in terms of communication with friends and family; social networks are used differently, depending on gender, when it comes to communicating with family and friends. This would be in line with previous studies [18,31], which also initially found that the use of social networks had led to detachment from family and friends, although it would be useful to explore gender differences.

It can be concluded that students' approaches to and use of social networks are different depending on the year of the university course, due to their increasing maturity. In this case, students made a different use of social networks in the more advanced years (sixth year) compared to the initial years (first year) in their university education process. Studies [51] have already shown that addiction to social networks decreases with age but is fuelled by the characteristics of physical and psychological immaturity of young people and adolescents, as well as their heavy use of them; a situation that can have more negative repercussions on them than on adults, hence the great interest in analyzing and reflecting upon it.

Another conclusion is that the area of study shows differences in the use of social networks, with engineering and social science students reporting more frequent use than

biomedical science students. However, biomedical and social science students seem to perceive social networks as more useful, compared to engineering students.

Age also appears as another variable to be considered in the use of social networks, as indicated by the significant differences between students according to age group. In this case, students in the 16–20 age group showed a higher expression of feelings in the use of social networks compared to other age groups. This could be due to the evolutionary development and constant use of social networks at an early age, a fact that is repeated in a similar way throughout different studies with similar characteristics [3,20,52].

Finally, and as a line of future research, it is recommended that the study be extended to other countries and contexts, incorporating qualitative methodologies that allow us to delve deeper into the reasons associated with the conclusions obtained in this study. It would also be interesting to conduct a longitudinal study with students, recording measurements at the beginning and end of the academic year, and analyzing the evolution of the degree of addiction, as well as its relationship with other variables.

As is well known, the instruments for diagnosing behaviour and actions based on the perceptions of the subjects themselves have the problem of not collecting direct information, but through the assessment made by the individuals themselves. The person may not be aware that they are performing such actions, or of the influence of certain phenomena and events.

**Author Contributions:** Conceptualization, C.L.C. and A.H.-M.; methodology, C.L.C., N.P.-C. and J.Q.-E.; investigation, C.L.C., A.H.-M., N.P.-C. and J.Q.-E.; writing-original draft preparation, C.L.C. and A.H.-M.; writing-review and editing, N.P.-C. and J.Q.-E. All authors have read and agreed to the published version of the manuscript.

**Funding:** The authors would like to thank that National University of San Agustín de Arequipa, UNSA, for funding the research project (contract number TP CS-06-2020-UNSA).

**Institutional Review Board Statement:** The study did not require ethical approval.

**Informed Consent Statement:** No personal data was collected and all the data in the survey was anonymised.

**Data Availability Statement:** The relevant data is primarily contained within the article.

**Conflicts of Interest:** The authors declare no conflict of interest.

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
