# Peer review of "Use of Social Networks in University Studies: A Peruvian Case Study"

_education, doi:10.3390/educsci12120893_

Round 1
Reviewer 1 Report
The article presents a very interesting study on the use of social networks and their effects on the university population. After reviewing it in depth, a series of recommendations are made for the authors to take into consideration.
1. The phrase "List three to ten pertinent keywords 17 specific to the article yet reasonably common within the subject discipline.)" in keywords should be deleted
2. Line 158-161 should be written as a paragraph
3. All figures and tables must be indicated in text
4.In the results writing, it is not necessary to copy the complete item writing, it can be referenced with Item 23, Item 1...
5. Use of Mann-Whitney U its correct but it should be justified and indicated in texto why authors use non parametric tests (non normality?, groups with few participants...?).
6. Same for Kruskall Walli tests. As with the anova, this type of test does not detect between which groups differences are obtained. Although it is indicated in the text, the results must be shown in another table or visually. In this case, its recommeded a graphic like this (https://i.stack.imgur.com/EobhW.png)
7. Some new references are recommended:
https://www.mdpi.com/955676
https://redined.educacion.gob.es/xmlui/handle/11162/199508
Author Response
REVIEWER 1
1. The phrase "List three to ten pertinent keywords 17 specific to the article yet reasonably common within the subject discipline.)" in keywords should be deleted.
Deleted sentence
2. Line 158-161 should be written as a paragraph
The sentence has been restructured 3. All figures and tables must be indicated in text
Modified 4.In the results writing, it is not necessary to copy the complete item writing, it can be referenced with Item 23, Item 1...
Thank you for your consideration, but we think it is more explicit for the reader to know which item is being referred to. 5. Use of Mann-Whitney U its correct but it should be justified and indicated in texto why authors use non parametric tests (non normality?, groups with few participants...?).
The the following text has been added to the manuscript: “It has been verified that the data are not normally distributed through a descriptive study in which skewness and kurtosis have been taken into account. The Kolmogorov–Smirnov goodness-of-fit test confirmed this finding, with a significance (p-value) equal to 0.000 for all items (non-normal distribution)” 6. Same for Kruskall Walli tests. As with the anova, this type of test does not detect between which groups differences are obtained. Although it is indicated in the text, the results must be shown in another table or visually.
The average range analysis table has been omitted as it is too large. For this reason, the most significant aspects of the table have been commented on. 7. Some new references are recommended: https://www.mdpi.com/955676 https://redined.educacion.gob.es/xmlui/handle/11162/199508
Added suggested reference

Reviewer 2 Report
Dear authors,
Your work is interesting, but I think it needs some changes, that I suggest in the attached file.

Round 2
Reviewer 2 Report
I am sorry, but the authors have made minimal changes. From my point of view the paper is still flawed. In addition, they have not responded to or fixed some of the suggestions from the first revision.
Abstract
They have added the suggestion, without modifying the previous one. Now it is too long.
Introduction
A restructuring was recommended, the only thing they have done is to remove the title "Social networks".
Materials and methods
Reference 41 is misspelled.
In the results section some things have been fixed, but others have not been resolved.
Discussion
Only the limitations have been added, although the wording of the text is not clear.
Author Response
Dear Reviewer,
following the comments from your second round, we have restructured the article, taking into account the suggestions made, as well as modifying numerical elements of the methodological part that were flawed.
We hope that it will now be considered for approval.
We thank you in advance for your feedback.
Please see attachment "cover letter round2" and modifications to the manuscript itself

Round 3
Reviewer 2 Report
No comments.